# OpenReview forum: "Multi-Session Budget Optimization for Forward Auction-based Federated Learning"
_ICLR.cc/2025/Conference — Submitted to ICLR 2025_

### Official Review · Reviewer_k8dF · 2024-10-31

**Soundness:** 2
**Presentation:** 1
**Contribution:** 2
**Rating:** 3
**Confidence:** 3

**Summary:**

The work propose MultiBOS-AFL, a budget optimization strategy for auction-based Federated Learning that enables data consumers (DCs) to strategically allocate budgets over multiple training sessions. Using hierarchical reinforcement learning, MultiBOS-AFL optimizes both session-level budget pacing and within-session bidding, allowing DCs to maximize data acquisition and model utility within budget limits.

**Strengths:**

- MultiBOS-AFL is the first of its kind to implement budget pacing in multi-session auction-based federated learning.

- Experimental results demonstrate consistent outperformance over state-of-the-art approaches, achieving higher data acquisition rates and improved FL model accuracy.

- Leveraging hierarchical reinforcement learning allows MultiBOS-AFL to adjust its bidding strategies dynamically, optimizing DC outcomes in real-time.

**Weaknesses:**

- The explanation of the DOs' Reputation Calculation is unclear. The paper lacks motivation for incorporating the Beta Reputation System to compute the reputation value $v_i$. It is unclear how the Shapley Value metric relates to the Beta Reputation System, and the references provided do not adequately support this rationale. Please clarify the rationale behind combining the Shapley Value and Beta Reputation System and explain why these methods complement each other in this context.


- The Shapley Value-based contribution metric depends on the post-trained model using DO $i$'s data, $f(W\_{S\cup\{i\}})$, which is unrealistic. Before deciding whether DO $i$ will be accepted for training, we cannot access or use its data to obtain a post-trained model and then evaluate the model's performance. The reviewer suggests clearly explain how  to estimate the Shapley Value without prior access to the DO's data.

- The proposed multiple auction-based strategy has a weak connection to the federated learning training process. It is unclear how the selection of DOs impacts the overall FL model performance in practice.

**Questions:**

- Typo in line 189: "It is important to highlight that, as depicted in Eq. equation  3."

- Why Beta Reputation System can be used to compute the reputation value of DO $i$? What is the rationale behind its use?

- Except for the total budget, what is the connection between independent multiple sessions?

- If there are any approximation methods or alternative metrics that could be used instead of the Shapley Value?

---

### Official Review · Reviewer_B9oy · 2024-11-03

**Soundness:** 2
**Presentation:** 3
**Contribution:** 2
**Rating:** 3
**Confidence:** 4

**Summary:**

This paper introduces an RL-based approach for auctions in federated learning. The goal is to maximize the total "reputation" of data owners that are won by the data consumer (who aims to train a model via FL) across multiple training sessions under a budget constraint. To this end, the RL approach consists of two RL agents, inter-session budget pacing agent (InterBPA) and intra-session budget management agent (IntraBMA). These two agents learn to allocate the budget across different sessions, and to determine the bid price for each data owner under the session budget in each session. The two agents are trained with DQN techniques.

**Strengths:**

S1. A well-motivated problem is studied.

S2. The AFL scenario considered in this work is practical.

S3. The approach is presented in a clear way.

**Weaknesses:**

W1. Some settings and assumptions of the task need to be explained (see questions below).

W2. In a competitive environment for DO auctions, it is unclear why DCs will follow the same approach (e.g., the one proposed in this paper). Any analysis about the incentives or truthfulness?

**Questions:**

Q1. Some settings and assumptions of the task need to be explained. I understand that some settings of this paper follow previous works. But more background about these settings is needed in Section 3. Also, it does not mean that all the settings are reasonable. Some of them need to be motivated better: for example,
- It is unclear how (2) and (3) can be evaluated efficiently across the FL training process to derive the reputation $v^i_s$ of each DO in each session. If they cannot be evaluated exactly, any approximation? How does the approximation affect the reward and in turn the quality of the learned model?
- What is the auction mechanism used in this work? This mechanism is supposed to affect the learned policies in different agents.

Q2. In a competitive environment for DO auctions, it is unclear why DCs will follow the same approach (e.g., the one proposed in this paper). Any analysis about the incentives or truthfulness?
- Are different DCs forced to use the same RL approach? In a competitive market, it is not realistic unless all DCs have sufficient incentive to do so (e.g., under a truthful mechanism).
- Following the second question in Q1, how do different auction mechanisms affect the learned policies? Experiments can be conducted for two or three such mechanism (e.g., first-price/second price)

Q3. The presentation is clear in general, some minor things to be fixed, e.g., $S$ appears twice in (1) and (2) - I know the fonts are different but it is still a bit confusing.

---

### Official Review · Reviewer_kDTh · 2024-11-06

**Soundness:** 2
**Presentation:** 3
**Contribution:** 2
**Rating:** 5
**Confidence:** 2

**Summary:**

The paper presents MultiBOS-AFL, a strategy for budget optimization in forward auction-based federated learning (AFL). This work extends data owner recruitment from single session to multi-session AFL, optimizing budget pacing over time. This approach features an inter-session budget pacing agent (InterBPA) and an intra-session bidding agent (IntraBMA), which jointly allocate and bid over multiple auction sessions.

**Strengths:**

- The paper tries to look into a new multi-session AFL scenario, expanding budget optimization strategies to cover temporal dimensions.

- paper is well written and easy to follow

**Weaknesses:**

- Applicability to real-world scenarios: The paper’s experimental settings use controlled noise and IID versus non-IID setups, but do not test more realistic settings with fluctuating budgets or DO availability patterns. In the real world, the DO availability is unknown, how does the proposed method handle that?

- How realistic is the proposed scenario of a multi-session auction? More motivation of why this scenario is valid would be helpful.

- In the evaluation section, this work is not compared to other existing HRL algorithms. It is important to compare to existing HRL algorithms and show its performance benefit.

**Questions:**

- Adaptability in Dynamic Environments: How would MultiBOS-AFL perform if DOs arrived dynamically rather than being preset in each session? Could the HRL agents adapt if the number of sessions varied unexpectedly?

- Computational Efficiency: What is the computational overhead associated with HRL training, especially as the number of sessions and DOs increase?

- Can the proposed method scale to a larger dataset such as imagenet? Is there any limitation?

- Can the proposed method generalize when the demand or supply changes over time? It would be helpful to provide some analysis.

---

### Official Review · Reviewer_YgWp · 2024-11-09

**Soundness:** 3
**Presentation:** 3
**Contribution:** 3
**Rating:** 6
**Confidence:** 2

**Summary:**

Auction-based federated learning consists of three types of participant: 1) data owners, who are willing to share their potentially sensitive data if they are appropriately compensated; 2) data consumers, who need data in order to train their federated learning models; and 3) a trusted third-party “auctioneer” who orchestrates the auction between data owners and data consumers. In scenarios where data consumers compete for data from the same pool of data owners, the key challenge for data consumers is to develop an optimal bidding strategy that maximizes certain key performance indicators subject to a budget. Previous work on optimizing this objective assumes that there is a single auctioning session, but in practice data consumers might recruit data owners over multiple training session. This paper proposes a budget optimization strategy for multi-session auction-based federated learning. The budget optimization strategy is based on hierarchical reinforcement learning with two types of budget allocation agents to handle the budget allocation both during a session and between sessions.

**Strengths:**

* The empirical evaluation is thorough, includes a decent number of datasets and also demonstrates good performance compared to previous methods.
* Extending budget optimization strategies to the multi-session AFL setting seems like an interesting and practical problem.

**Weaknesses:**

* The paper is dense with technical jargon. (One easy improvement could be to use acronyms less frequently, e.g. limit it to a max of one acronym per sentence).
* It is hard to get a sense of the novelty of the proposed method and how much of a jump it is from previous work. I would have also liked to see a more thorough discussion on the techincal challenges in extending budget optimization strategies to the multi-session AFL setting.

**Questions:**

Looking at footnote 2: “maximizing the total utiltiy is equivalent to optimizing the performance of the global FL model obtained by the target DC.” What is the distinction between “utility” and “test accuracy” as metrics?

---

### Meta-Review · Area_Chair_bd2q · 2024-12-19

**Metareview:**

The paper studies an auction-based FL setting, particularly one in which auctions span multiple sequential sessions. An RL algorithm is proposed to jointly optimize budget pacing and bidding across sessions, and evaluated experimentally.

Strengths included the thorough experimental evaluation and the importance of the problem studied. Some reviewers found the presentation dense and key concepts unclear, and others were concerned with the motivation, particularly whether rational participants would engage in the scheme proposed. The contribution appears to be algorithmic and, as an application of hierarchical FL, not that novel; issues of incentive compatibility, truthfulness, etc. that are present in auction literature are largely ignored. Issues of complexity are also treated superficially.

**Additional Comments On Reviewer Discussion:**

Reviewers remained concerned about the overall motivation of simply studying this from a utility optimization perspective, ignoring other often encountered auction desiderata, the treatment of complexity, and the overall novelty.

---

### Decision · Program_Chairs · 2025-01-22

Reject